# Relationship between Diet, Microbiota, and Healthy Aging

**DOI:** 10.3390/biomedicines8080287

**Published:** 2020-08-14

**Authors:** Elisa Sanchez-Morate, Lucia Gimeno-Mallench, Kristine Stromsnes, Jorge Sanz-Ros, Aurora Román-Domínguez, Sergi Parejo-Pedrajas, Marta Inglés, Gloria Olaso, Juan Gambini, Cristina Mas-Bargues

**Affiliations:** 1Freshage Research Group, Department of Physiology, Faculty of Medicine, University of Valencia, CIBERFES-ISCIII, INCLIVA, 46010 Valencia, Spain; isanes@alumni.uv.es (E.S.-M.); lucia.gimeno@uv.es (L.G.-M.); krisbaks@alumni.uv.es (K.S.); sanzros@alumni.uv.es (J.S.-R.); aurodo@alumni.uv.es (A.R.-D.); sergi_spp_95@hotmail.com (S.P.-P.); gloria.olaso@uv.es (G.O.); cristina.mas@uv.es (C.M.-B.); 2Department of Biomedical Sciences, Faculty of Health Sciences, Cardenal Herrera CEU University, 46115 Valencia, Spain; 3Freshage Research Group, Department of Physiotherapy, Faculty of Physiotherapy, University of Valencia, CIBERFES-ISCIII, INCLIVA, 46010 Valencia, Spain; marta.ingles@uv.es

**Keywords:** Mediterranean diet, Oriental diet, nutrition, polyphenols, microbiota, aging, health

## Abstract

Due to medical advances and lifestyle changes, population life expectancy has increased. For this reason, it is important to achieve healthy aging by reducing the risk factors causing damage and pathologies associated with age. Through nutrition, one of the pillars of health, we are able to modify these factors through modulation of the intestinal microbiota. The Mediterranean and Oriental diets are proof of this, as well as the components present in them, such as fiber and polyphenols. These generate beneficial effects on the body thanks, in part, to their interaction with intestinal bacteria. Likewise, the low consumption of products with high fat content favors the state of the microbiota, contributing to the maintenance of good health.

## 1. Introduction

One of the most significant changes in today’s society is that the population is aging more slowly than in the past. We live longer; the majority of the population has a life expectancy of 60 years or more. According to data from the World Health Organization (WHO), the percentage of people over 60 years will be duplicated on a worldwide scale before 2050 [1]. In 2020 the number of people of more than 60 years of age has been reported to be higher than those under five years, and by 2050 the number of elderly adults requiring help with daily tasks will quadruplicate [2].

Advances in medicine and public health care, as well as lifestyle changes, have had positive effects on life expectancy, mortality rates, and chronic disease prevalence, thereby causing an aging of the population [3].

Whether or not the impact of these factors on society is positive depends on the health status of the elderly. Therefore, it is important to prevent diseases associated with age, such as diabetes and cardiovascular diseases, to promote an active aging of the population and maintain social, mental, and physical wellbeing. In this regard, three pillars of active aging have been postulated: Participation, health and security [4,5]. We will focus our study around health.

It is important to keep in mind that the risk factors causing age related diseases can be modified. One of the factors that can act upon the health pillar is nutrition [6]. What, when, and how much we eat affects the quality of life. A well-balanced diet can modulate the proliferation of specific bacteria within the gut microbiota, which has been related to an improved health status for the elderly.

### Microbiota

In this century, research related to microbiota has increased due to two primary advances: The holobiont theory of evolution, and metagenomics. The former one plays an important role in the physiology of superior organisms, and the latter one allows for the identification of microorganisms without a need for cultivation. As a result of these advances and of the increasing interest in this field of study, the concept of the “intestinal–brain axis” was created, with a tight relation between the intestinal microbiota and neurodegenerative diseases such as Parkinson, Alzheimer, stroke, and epilepsy [7]. Actually, a study performed in mice proved the presence of bacteria in the brain of these mice [8]. Furthermore, Roberts et al. obtained a micrography of a human brain where bacteria were visible in all the cerebral blood vessels, proving the existence of a brain microbiota [9]. Further supporting the idea of the presence of bacteria in the brain, a study using epilepsy patients demonstrated a reduction of convulsion frequency by antibiotic treatment [10]. All these data suggest that there might be a brain microbiota, and by extrapolation a whole organism microbiota.

The gastrointestinal tract is colonized by a set of microorganisms that include not only bacteria but also viruses, fungi, and protozoa. Unlike other microorganisms, these are not identified as pathogens by our immune system, but rather coexist symbiotically with the enterocytes [11]. So far, it is known that its composition contains a total of 52 different phyla and up to 35,000 different bacterial species [12], the large majority being Firmicutes, Bacteroidetes, Actinobacteria, and Proteobacteria (Figure 1).

It is worth mentioning that there are differences in composition and number depending on the location along the gastrointestinal tract such that at the stomach level they are estimated at a density of 10^2^ bacteria/mL and in the colon at around 10^11^ bacteria/mL [14].

Originally, the intestinal microbiota is formed in the placenta, where there are low levels of non-pathogenic bacteria, the majority being *Firmicutes* and *Bacteroidetes*. After birth, the child’s intestine becomes rapidly colonized by the microbiota. Several studies show that this could already happen at the level of the uterus. Depending on whether the delivery is vaginal or by caesarean section, the intestine of the infant will be colonized by organisms from the maternal vagina or from the maternal skin flora. Another factor that influences the acquired microorganism is the type of diet the infant receives, breast milk or formula foods [11]. During the first three years of life, the child’s microbiota is unstable and has little diversity. After the third year it acquires a milieu similar to that of the adult stage. In the case of older adults (>65 years), the microbiota is characterized by a decrease in Firmicutes and Bifidobacterium, with diversity in the patterns of abundance for Clostridium [15]. In relation to centenarians, differences in the composition of the microbiota are also evident, especially at the level of the Firmicutes subgroups, with a decrease in Clostridium and an increase in Bacilli. In addition, it is enriched by the edge Proteobacteria. Thus, as we age, progressive changes are produced in the morphology and function of the microbiota. This could alter its diversity and provoke inflammatory and metabolic disruption, causing inflammatory diseases in the intestine, such as irritable bowel, obesity, etc. [16]. The relation Firmicutes/Bacteroidetes is altered when the Firmicutes proportion is augmented over Bacteroides and generates consequences such as those found in obesity. However, this association is still polemic and under investigation [17,18]. While, on a global level, the Firmicutes and Bacteroidetes seem to decrease, Clostridium is increased in elderly adults [19]. In the last years, bacteria of the Clostridium genera have provoked a big interest for their impact on morbidity and mortality in the older generation [20]. They can act as opportunistic pathogens and cause diseases like intoxication through alimentation, necrotizing enterocolitis, or necrotic enteritis [21]. Furthermore, their incidence is increasing, and they have become the most frequent cause for nosocomial diarrhea, which can lead to the production of toxic megacolon and sepsis, and even to death [22].

However, through nutrition we can modify the microbiota to maintain intestinal health by stimulating beneficial bacteria, such as the Bacteroidetes and Firmicutes in balanced proportions, and diminishing prejudicial bacteria like Clostridium [16,23]. Additionally, maintaining a healthy diet throughout a lifespan might be beneficial for healthy aging.

## 2. Interplay between Aging and Microbiota

The composition and function of the microbiota changes in individuals of advanced age (>65 years), as has been mentioned earlier. Between physiological changes and lifestyles associated with aging, we find a decrease in dentures, a reduction in digestion and absorption, changes in appetite due to medication, and changes in living conditions such as hospitalization or elderly residences [24]. These factors might be responsible for changes in alimentation habits, and thereby nutrition.

The mechanism by which the microbiota changes with age is not yet fully understood. Lifestyle changes, and particularly diet, play an important role. As aging is often accompanied by a reduction in quantity and variety of aliments containing fiber, it often leads to a risk of malnourishment [25].

Additionally, the microbiota can modulate changes in aging related to innate immunity, sarcopenia, and cognitive function, which are essential components of the frailty syndrome [26]. This syndrome, was defined for the first time by Linda Fried in 2004 as “the physiological state characterized by an increase in vulnerability to external aggressors, as the result of a decrease or deregulation of the physiological reserves of multiple systems, which causes difficulties in maintaining homeostasis” [27]. Recent studies have suggested that the loss of intestinal microbiota is more related to age associated frailty than to chronological age [28]. In this context, elderly people can experience comorbidity affecting the intestine and its bacteria.

In 2007, a study performed by the ELDERMET group in Cork, Ireland, proved the existence of a correlation between diet, the microbiota, and health in elderly volunteers. The microbiota in relation to aging was studied in both community and long-stay residential care elderly subjects. The first group showed a loss in bacterial diversity due to a higher intake of medication, and a recuperation of microbial diversity when reducing antibiotics, whereas the second group showed a loss in microbial components associated with bad health, and a gain in microbiota associated with aging. Therefore, the previous correlation can be used to measure the difference between elderly that live in elderly homes or rehabilitation centers and those who live by themselves in the long run [29]. Claesson et al. supported these findings on the relation between diet, microbiota, and health status. Through analysis of separated fecal microbiota composition, they showed that in 178 subjects of advanced age a change in diet associated with that found when moving to a care facility generated a change in the composition of intestinal bacteria significantly correlated with measures of frailty, comorbidity, nutritional status, and inflammatory markers [15].

The most noticeable changes in the intestinal microbiota are produced during the transition from adult to elderly. In comparison to younger individuals, a decrease in microbial diversity is observed, specifically in Bifidobacterium, Firmicutes, and Clostridium [15]. These changes in the microbiota associated to age are a factor of high relevance in several diseases, such as chronic inflammation, neurodegeneration, cognitive deterioration, frailty, and diabetes types 1 and 2 [30]. In regard to extreme longevity, the microbiota of centenarians differs from that of elderly adults. Bacteroidetes and Firmicutes dominate their intestinal microbiota, making up more than 93% of the overall bacteria flora. However, comparing to younger adults, specific changes in the subgroups of Firmicutes are observed: Clostridium is decreased, and Bacilli augmented. Moreover, the intestinal microbiota of centenarians in enriched in Proteobacteria, a group containing bacteria recently redefined as “pathobionts”, benign endogenous microbes which in dysbiosis can lead to pathologies [31,32].

Taken together, intestinal microbiota is subjected to constant changes throughout life and continues to change in old age and is closely linked to diet and health status, suggesting the possibility that microbial profiling may serve as a clinically useful biomarker in geriatric care. Predomination of one type of bacteria, or the over proliferation of bad bacteria upon good bacteria will have negative consequences. Thus, maintaining intestinal microbiota diversity seems to be a key point when seeking geriatric health [33]. Figure 2 shows microbiota evolution through aging.

## 3. The Influence of Nutrition on the Microbiota and Aging

In this paragraph, we will comment on the two diets that are the most associated with healthy aging: the Mediterranean and the Oriental diets. 

### 3.1. The Mediterranean Diet

The Mediterranean diet (MD) is characterized by a high daily consumption of fruits and vegetables, whole grains and legumes, and a lower consumption of meat, fish, and lactose-rich products. Additionally, it is also associated with olive oil as the major source of fats, and moderate intake of red wine [34,35,36].

It has been shown that following an MD can have multiple benefits as it delays the appearance of chronic diseases associated with age such as cancer, diabetes type 2, and neurodegenerative diseases [36,37,38]. Furthermore, recent data suggest that the MD promotes beneficial effects on the intestinal microbiota, favoring the diversity of the colon microbiota by increasing Bacteroidetes and Firmicutes and reducing Clostridium [37]. Another study carried out on 27 healthy subjects, who were monitored for MD compounds, showed how MD eating habits modified the composition and diversity of the microbiota. More specifically, they observed an increase in Bacteroidetes and a higher Bacteroidetes/Firmicutes ratio. This modification of the different microbiota populations has proven to exert anti-inflammatory effects. Accordingly, how MD can decrease the transcription factor E2F1 in colon cancer cells in order to decrease the proliferation of these cells has been studied. Additionally, it activates factors with protective effects against oxidative stress such as the transcription factor NRF2 [39,40].

Other studies support the beneficial effects of MD on cardiovascular diseases [39,41,42,43,44]. In one of these, low levels of triglycerides, high HDL cholesterol levels, and lower arterial pressures were found in those following the MD in comparison to a control group consuming a low-fat diet [43]. In the “Lyon Diet Study”, the risk of severe cardiovascular events was reduced by 76% in the MD group. Additionally, a multicenter randomized study in Spain obtained similar results [44]. Moreover, how the risk of several types of cancer is decreased has also been studied, although the underlying mechanisms are still not clear [45,46]. This is reflected in the 605 patient cohort participating in the Lyon Heart Study, where the cancer risk decreased by 61% in the MD group in comparison to the control group [47]. One of the components of the MD is hydroxityrosol, a compound present in olive oil with antioxidant properties. It has been demonstrated that it reduces proliferation and induces apoptosis of certain cancer cells in the colon [48,49], prostate [50], and pancreas [51]. Furthermore, a clinical trial performed in Spain showed that the combination of the nutrients related to MD reduced pain and the inflammatory marker C-reactive in patients with breast cancer [52,53]. Likewise, various studies have analyzed the correlation between MD and diabetes type 2. MD has been seen to attenuate the augmentation of postprandial glycemia, reduce high peaks of insulin secretion, as well as diminish hyperlipidemia in patients with type 2 diabetes [54]. The PREDIMEM study “Prevention with the Mediterranean Diet” found that a bigger adherence to MD was inversely associated with the incidence of type 2 diabetes, which was reduced by 52% in the MD group compared to the control group [55]. In relation to neurodegenerative diseases, a high adherence to MD has been associated with improved cognitive function and lower risk of dementia [56,57,58,59]. However, many of these studies are observational, and can therefore not display a cause–effect relation. Moreover, other studies confirm the improvement in cognitive function in comparison to low-fat diet groups [47,60,61]. 

As we have previously described, MD reduces the risk of several diseases associated with aging. Furthermore, many studies demonstrate that MD promotes changes in the intestinal microbiota associated with healthy beneficial bacteria [34,37,39]. Consequently, the manipulation of the microbiota through MD should improve or benefit the treatment of these diseases [62,63,64,65]. However, the mechanisms through which MD promotes healthy changes on the intestinal microbiota are still not completely known. 

### 3.2. The Oriental Diet

The Oriental Diet (OD) is well known for its health benefits. It has been demonstrated that the population following this diet has a longer lifespan, to the point where Japan has become one of the countries with the highest life expectancy rate. In addition to the long lifespan, the diet is also associated with a long health span. In the OD, one of the most common aliments is soy. 

It is known that soy consumption has positive effects on the organism due to its ability to increase intestinal bacteria diversity. Tamura et al. divided rats in three groups based on diet for a two week period. The different diets were enriched either with casein protein, soy protein, or soy protein and high fat. After studying the microbiota through genes of 16S rRNA of feces, they were able to see how soy protein induced a bigger variety in intestinal bacteria, which is associated with a healthy microflora. Specifically, they observed an increase in Bacteroidetes and Proteobacteria, as well as in Bifidobacterium and Enterococcus. On the other hand, they also addressed a decrease in Firmicutes and Lactococcus [66].

Soy contains a variety of polyphenols, genistein being one of them. Of all groups of polyphenols (stilbene, ligand, alcoholic polyphenols, etc.), genistein is an isoflavone, the structure of which is similar to that of estrogen, and is therefore also classified as a phytoestrogen [67]. Genistein is found almost exclusively in legumes, such as chickpeas and peas, and majorly in soy and its derivatives [68].

It has been observed that genistein consumption can have multiple benefits, within these the promotion of cardiovascular health, and thereby flexibility of blood vessels and reduction in cholesterol concentration [69]. As with many polyphenols, it is transformed and modified by the microbiota to enable its actions in the organism. It interacts with estrogenic receptors to alleviate symptoms associated with menopause, hot flashes, and osteoporosis, as well as exerting antiaromatase activity, which is key in the protection against breast cancer [70]. Several studies performed on rats have shown how the administration of genistein at early age counteracts dysbiosis in the intestinal microbiota provoked by a high fat diet [71]. These data are supported by research by Zhou et al. where they observed beneficial bacteria proliferation through perinatal supplementation [72]. López et al. also found that genistein can regulate the intestinal microbiota by reducing metabolic endotoxemia and the neuroinflammatory response to a high fat diet [73]. Additionally, genistein has antioxidative effects, as it blocks the formation of free radicals, hydrogen peroxide, and superoxide anion, and by overexpressing antioxidant enzymes [74]. Through these effects, it also has protective effects against ischemic stroke and neuroprotective effects against Alzheimer Disease by reducing the hyperphosphorylation of Tau protein [75,76]. Lastly, it has anti-inflammatory and antitumoral effects through its ability to inhibit key enzymes in the appearance and progression of tumors. It also regulates the tumoral microenvironment, causing a higher sensitivity to therapies by modulating the production of cytokines and chemokines, and activating immune cells [77,78].

Another polyphenol present in soy is daidzein; similarly to genistein, it is modified and transformed by the microbiota to equol, a compound with anti-inflammatory and anticarcinogenic properties. This has been demonstrated in several studies where it has been shown how daidzein inhibits the production of IL-2 and how the proliferation of tumoral cells in breast cancer is regulated by the blockage of PI3K/AKT [79]. Furthermore, it has been determined that equol protects β-pancreatic cells against streptozotocin-induced diabetes in rats [80]. 

Curcumin, a polyphenol found in turmeric root, is also abundantly present in OD. It has proven to be capable of exerting anti-inflammatory and anti-tumor effects, among others [81,82]. Regarding its role in the microbiota, it has been seen to directly modify it. Since its absorption and bioavailability is low, it can be found in significant amounts in the gastrointestinal tract after its consumption. In fact, curcumin administration considerably changed the relationship between beneficial and pathogenic bacteria by increasing the abundance of Bifidobacteria and Lactobacili and reducing the loads of Prevotellaceae, Choriobacteria, Enterobacteriaceae, and Enterococci [83].

Thereby, the OD reduces the risk of several diseases associated with aging. Additionally, numerous studies have shown how following an OD has positive effects on the microbiota by increasing the number of beneficial bacteria. However, as with the MD, the mechanisms by which this diet promotes healthy changes in the intestinal microbiota are still not fully known. 

### 3.3. Effects on the Microbiota by Compounds Present in Diet: Fiber, Fats, and Polyphenols

As we have previously described, Mediterranean and Oriental diets have modulatory effects on intestinal microbiota. These diets include a wide variety of nutrients in balanced proportions that promote gut bacteria diversification and thus a healthy microflora. While practically all compounds we ingest through diet can have effects on the microbiota, in this review we will only focus on food compounds that are similar in the two diets, which include a high consumption of polyphenols and fibers, and a higher proportion of unsaturated fats (Table 1).

In the upcoming paragraphs, we will discuss in depth the beneficial effects of each one of these compounds.

#### 3.3.1. Fibers

Diets rich in fibers such as, β-glucan, arabinoxylan, galactomannan, and pectines facilitate weight control, decrease systemic inflammation, and are fundamental in the maintenance of intestinal health by supplementing substrates for the health promoting bacteria (Bifidobacterium and Lactobacillus) [84]. 

Studies have shown that a diet rich in fiber generates beneficial effects for elderly adults, especially on the cardiovascular system. Fiber-rich diets reduce the risks for cardiovascular disease, such as obesity, diabetes, and hypertension. The inverse correlation between high consumption of fiber and weight gain risk has been demonstrated [38]. Additionally, fiber seems to have a positive impact on the composition of the intestinal microbiota, augmenting the number of beneficial bacteria, inhibiting the growth of pathogens, and reducing atherogenic serum cholesterol in the microbiome. Furthermore, it prohibits the intolerance to glucose by reducing postprandial hyperglycemia through the formation of a viscous layer around the small intestine, thereby slowing down the chyme transition. This in turn augments the thickness of the aqueous layer where solutes must pass to reach the enterocytic membrane, causing a decrease in glucose, lipids, and amino acid absorption. As a result, less biliary acids are absorbed, and in turn cholesterol levels are decreased, which employs the synthesis of new biliary acids [85]. 

In this manner, a high consumption of soluble and insoluble fibers is associated with lower risk of stroke, as shown in a study performed over 12 years [86]. Another study demonstrated that the ingestion of total fiber, cereal fiber, and vegetable fiber was associated with a lower risk of hypertension [87]. Not only that, a study about the association between ingested fiber in the diet per 1000 kcal and leucocyte telomere length showed that adults consuming high quantities of fiber had longer telomeres, which suggests that biological aging was significatively lower [88]. 

In conclusion, a diet rich in fiber facilitates the presence of beneficial intestinal bacteria, and thereby lowers the risk of diseases in elderly adults, leading to healthy aging. 

#### 3.3.2. Fats

Classic epidemiologic studies have demonstrated the relation between diets with a high content of saturated fats, and a higher risk of age-related diseases [89,90]. 

Studies with animals have shown that a high fat diet can determine the microbiome composition and affect the proportion of various families of bacteria in the intestine. In particular, rats that received a high fat diet showed a lower number of Bacteroidetes and a higher number of Firmicutes and Proteobacteria, which is a characteristic of obesity [91]. This change in the microbiota composition could occur rapidly. For example, the change from a diet poor in saturated fats and rich in vegetal polysaccharides to a diet high in fats and sugars changes the structure of the microbiota in one single day. Additionally, a diet high in saturated fats is related to increased risk of chronic diseases, such as metabolic syndrome, hypertension, and atherosclerosis due to the alterations of the microbiota [92,93,94]. 

More importantly, if we compare humans with two different diets, it is clear that the microbiome diversity in our intestines is affected by what we eat. An example of this is the proportion of Bacteroidetes/Firmicutes which was much higher in African children compared to Italian, as they follow a diet of less fats and sugars [95]. This is further confirmed in a study by Wan et al. where they showed that healthy young adults consuming high quantities of fats had adverse effects on the intestinal microbiota, fecal metabolites, and plasmatic proinflammatory factors derived from the intestine, which could serve as potential factors to modify long term health [96]. 

In conclusion, it is clear that fiber has a protective effect in the elderly, especially at the cardiovascular level, and can contribute to healthier biological aging, whereas a diet high in fats induces a dysbiosis which can trigger metabolic alterations associated with chronic diseases and those related to age. Additionally, it has also been shown to have protective effects in younger individuals. 

#### 3.3.3. Polyphenols

Polyphenols, which have been mentioned earlier in this review, can reach the intestinal microbiota and thereby modify the molecule itself, the metabolism of the bacteria, and the intestinal flora population. This seems to have beneficial effects, including anticarcinogen, anti-inflammatory, and antioxidant properties [97]. 

For example, ellagitannins are polyphenols present in walnuts, pomegranates, and raspberries, which, when in the intestine, are modified by the microbiota and transformed into different compounds. Urolithin is one of the most widely studied products, and it has been demonstrated that it can be absorbed through enterohepatic circulation and that it can be transported in blood, and thereby be distributed to different tissues and exert diverse actions. It possesses anticancer effects through the inhibition of the Wnt signalization pathway, which could have a protective effect against colon cancer [98]. It has also been attributed anti-inflammatory properties as it inhibits the formation of proinflammatory compounds, such as cyclooxygenase-2 (COX-2), prostaglandin synthase E (PGSE), and prostaglandin E2 (PGE2) [99]. An important finding in this regard was made by Selma MV et al. in 2014, when they discovered a new species of bacteria in the microbiota responsible for the synthesis of urolithin from ellagic acid, which is formed in the stomach, that they denominated *Gordonibacter urolithinfaciens* [100]. However, the metabolites obtained from polyphenols through diet vary between individuals depending on the microbiota. In fact, certain individuals do not synthesize urolithin at all after the ingestion of its precursors [100]. 

Other polyphenols, such as resveratrol and curcumin, exert specific changes on the intestinal microflora. Concretely, in a study where rats were administrated these two polyphenols, they observed alterations in the bacteria groups Bacteroidetes and Clostridium. This in turn provided metabolic benefits in these individuals such as glycemic control [101].

On the other hand, there are polyphenols, such as quercetin, caffeic acid, and rutin, that affect the fermentation balance between the principal groups of intestinal bacteria that influence health by proliferating Bifidobacterium and diminishing Bacteroidetes and Firmicutes. Additionally, these polyphenols stimulate the production of short chain fatty acids through these bacteria [102]. 

These findings indicate that there are specific populations of bacteria capable of synthesizing final products such as urolithin and equol. The population of these bacteria in individuals is heterogenous, meaning the bacteria population present in the microbiota in each individual is a personal characteristic trait.

## 4. Conclusions

Life expectancy is increasing considerably, directing us towards a model of an aging society, and it is therefore necessary to expand our knowledge on the effects of aging. Taking into account all repercussions on the intestinal microbiota, to study the role of the microbiota is an interesting path, as well as the potential benefits that can be achieved through diet. Through dietary interventions many diseases associated with age such as cardiovascular and neurodegenerative diseases, diabetes, chronic inflammation, and cancer risk could be prevented. 

Therefore, it is important to continue researching different nutritional approaches to fight physiological damages that are produced in an organism by aging. In particular, chronic factors present in advanced stages of life, such as inflammation and oxidative stress, must be considered, as these factors can be diminished or augmented depending on the consumed diet. 

## Figures and Tables

**Figure 1 biomedicines-08-00287-f001:**
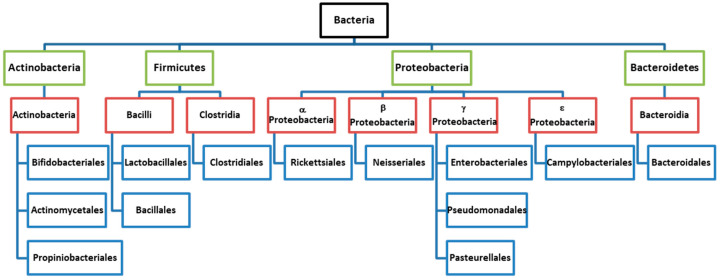
Taxonomy of microbiota. Black: Kingdom; green: Phylum; red: Class; blue: Order. Adapted from Unger et al., Pediatr. Res. 2015 [13]. Copyright © International Pediatric Research Foundation, Inc. 2015.

**Figure 2 biomedicines-08-00287-f002:**
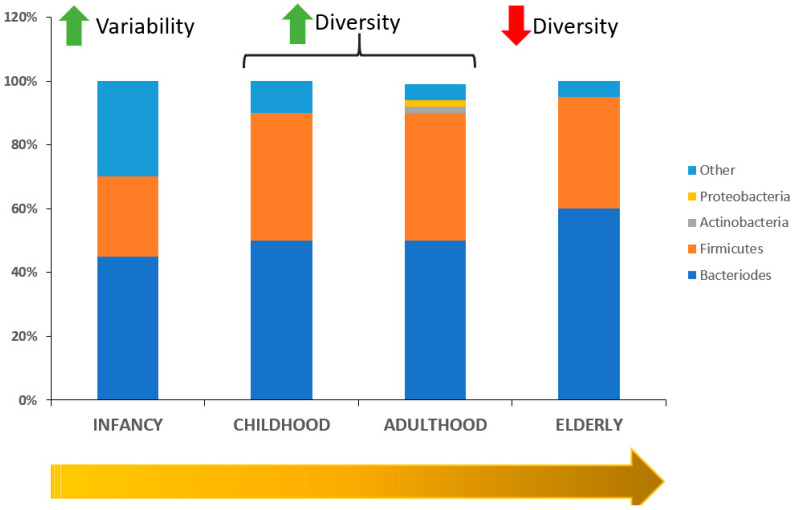
Intestinal microbiota changes through life.Green arrows signify increase. Red arrows signify decrease.

**Table 1 biomedicines-08-00287-t001:** Summaries of the main effects of each compound on the intestinal bacteria and the health consequences. Upward arrows signify increase. Downward arrows signify decrease.

Diet Compound	Microbiota Modification	Health Consequences
**Fibers**	↑Bifidobacterium ↑ Lactobacillus ↓ Pathogens	↓ Inflammation ↓ Hypertension ↓ Obesity
**Saturated fats**	↓ Bacteroides ↑ Firmicutes ↑ Proteobacteria	↑ Obesity ↑ Hypertension ↑ Atherosclerosis
**Polyphenols**	↑ Bifidobacterium ↓ Firmicutes ↓ Clostridium	↓ Inflammation ↑ Antioxidants

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
