# Peer review of "Relationship between Diet, Microbiota, and Healthy Aging"

_biomedicines, 2020, doi:10.3390/biomedicines8080287_

Round 1

Reviewer 1 Report

Comments to the authors

This is a well written and very interesting review on the role of dietary constituents on aging and microbiota.

Minor comments

  • Please revise the manuscript for appropriate use of English language (grammar and syntaxis).
  • Abstract Line 17 in the sentence “acting on the risk factors” please substitute the word "acting" with either "reducing, minimizing" or a similar word that states the “action” which is to lower the risk. Similarly, in lines 18-19 the sentence “Nutrition…allows us to modify these factors through modulation” is not correct use of English. Please rephrase
  • The figure below the (graphical) abstract does not appear correctly in the pdf format of the manuscript. Part of the figure and letters within, are covered/overlayed by another picture
  • Introduction Line 28 “the population is aging faster” should rather be corrected to “the population is aging more slowly”
  • Introduction Line 30 the abbreviated form “OMS” should also be written in full and passive voice should be used in the same sentence ie “the percentage of people over 60 will be duplicated”
  • Introduction Line 37 should be changed to” whether the impact of these factors on society is positive depends on….
  • In the microbiota section line 51, the authors introduce the concept of the intestinal-brain axis however, during the rest of the section the detailed focus is on intestinal microbiota. I would expect the addition of a brief paragraph on brain microbiota for reasons of emphasis (https://www.nature.com/articles/d41586-018-07416-8)(https://www.sciencemag.org/news/2018/11/do-gut-bacteria-make-second-home-our-brains)
  • The Role of Curcumin as a polyphenol in the Modulation of Ageing should be discussed within the section of oriental diet. Curcumin’s role in postponing ageing in animal models has already been documented (Bielak-Zmijewska A, Grabowska W, Ciolko A, et al. The Role of Curcumin in the Modulation of Ageing. Int J Mol Sci. 2019;20(5):1239. ) One of the roles of curcumin in the alleviation of ageing is reduction of inflammation (Skyvalidas DΝ, Mavropoulos A, Tsiogkas S, et al. Curcumin mediates attenuation of pro-inflammatory interferon γ and interleukin 17 cytokine responses in psoriatic disease, strengthening its role as a dietary immunosuppressant. Nutr Res. 2020;75:95-108. doi:10.1016/j.nutres.2020.01.005)

Author Response

RESPONSE TO REVIEWERS

 Resubmission of the manuscript with Reference ID biomedicines-890486

We thank the editor and the reviewers for their critical review and valuable comments. We have taken into account all of their recommendations and suggestions. Itemized responses are listed below citing line number and exact change. All the modifications have been clearly highlighted using the “track changes” function in Microsoft Word, so that changes are easily visible to the editors and reviewers (See Revised Manuscript-Tracked changes).

REVIEWER 1

Minor comments.

This is a well written and very interesting review on the role of dietary constituents on aging and microbiota.

  • Please revise the manuscript for appropriate use of English language (grammar and syntaxis).

ANSWER: As suggested by the reviewer, we made a careful revision of the manuscript checking for English language mistakes.

  • Abstract Line 17 in the sentence “acting on the risk factors” please substitute the word "acting" with either "reducing, minimizing" or a similar word that states the “action” which is to lower the risk. Similarly, in lines 18-19 the sentence “Nutrition…allows us to modify these factors through modulation” is not correct use of English. Please rephrase.

ANSWER: Following the reviewer recommendation, we changed both sentences. Please see the abstract on the new version of the manuscript with tracked changes.

  • The figure below the (graphical) abstract does not appear correctly in the pdf format of the manuscript. Part of the figure and letters within, are covered/overlayed by another picture.

ANSWER: We apologize, and we are sending again the graphical abstract figure in pdf format.

  • Introduction Line 28 “the population is aging faster” should rather be corrected to “the population is aging more slowly”

ANSWER: As suggested by the reviewer, we corrected the sentence. Please see the new version of the manuscript with tracked changes, line 36.

  • Introduction Line 30 the abbreviated form “OMS” should also be written in full and passive voice should be used in the same sentence ie “the percentage of people over 60 will be duplicated”

ANSWER: We apologize for our carelessness and we corrected the sentence using passive voice. Please see the new version of the manuscript with tracked changes, lines 38-39.

  • Introduction Line 37 should be changed to” whether the impact of these factors on society is positive depends on….

ANSWER: We thank the reviewer and modified the sentence as suggested. Please see the new version of the manuscript with tracked changes, line 45.

  • In the microbiota section line 51, the authors introduce the concept of the intestinal-brain axis however, during the rest of the section the detailed focus is on intestinal microbiota. I would expect the addition of a brief paragraph on brain microbiota for reasons of emphasis (https://www.nature.com/articles/d41586-018-07416-8) (https://www.sciencemag.org/news/2018/11/do-gut-bacteria-make-second-home-our-brains)

ANSWER: We appreciate the reviewer comment and we completed this section with the following paragraph and references in order to explain the brain microbiota. Also, please see the new version of the manuscript with tracked changes, lines 61-66.

“Actually, a study performed in mice proved the presence of bacteria in the brain of these mice [8]. Furthermore, Roberts et al., obtained a micrography of a human brain where bacteria were visible in all the cerebral blood vessels, proving the existence of a brain microbiota [9]. Further supporting the idea of the presence of bacteria in brain, a study using epilepsy patients demonstrated a reduction of convulsion frequency by antibiotic treatment [10]. All these data suggest that there might be a brain microbiota, and by extrapolation, a whole organism microbiota.”

  • The Role of Curcumin as a polyphenol in the Modulation of Ageing should be discussed within the section of oriental diet. Curcumin’s role in postponing ageing in animal models has already been documented (Bielak-Zmijewska A, Grabowska W, Ciolko A, et al. The Role of Curcumin in the Modulation of Ageing. Int J Mol Sci. 2019;20(5):1239) One of the roles of curcumin in the alleviation of ageing is reduction of inflammation (Skyvalidas DΝ, Mavropoulos A, Tsiogkas S, et al. Curcumin mediates attenuation of pro-inflammatory interferon γ and interleukin 17 cytokine responses in psoriatic disease, strengthening its role as a dietary immunosuppressant. Nutr Res. 2020;75:95-108. doi:10.1016/j.nutres.2020.01.005)

ANSWER: We took in consideration the reviewer suggestion and we added a paragraph about curcumin within the section of Oriental Diet. Please see the new version of the manuscript with tracked changes, lines 289-295.

“Curcumin, a polyphenol found in turmeric root, is also abundantly present in OD. It has proven to be able of exerting anti-inflammatory and anti-tumor effects, among others [81,82]. Regarding its role in the microbiota, it has been seen to directly modify it. Since its absorption and bioavailability is low, it can be found in significant amounts in the gastrointestinal tract after its consumption. In fact, curcumin administration considerably changed the relationship between beneficial and pathogenic bacteria by increasing the abundance of Bifidobacteria, Lactobacili and reducing the loads of Prevotellaceae, Choriobacteria, Enterobacteriaceae and Enterococci [83].”

Reviewer 2 Report

This is a very interesting review of an intriguing and very exciting problem of health related to the relationships existing between some components of the diets and the secondary effects on the intestinal microbiota in relation to the aging process.

I have included some small changes and suggestions in order to improve the manuscript

The references are very appropriate and in a good number included.

1.- I suggest a little change on your title by this other:

“ Effect of some diet components and intestinal microbiota composition on the achievement  and maintenance of good healthy aging”

2.- The Abstract, is OK

3.- The Introduction, is OK

4.- 1.1 Microbiota . , is OK

5.-2. The interplay between aging and microbiota. Is OK

Figure 2 : In the different sections of the circle  components of different microbacteria populations, would be illustrative to include  the percentage number of every population in the diverse periods of life  shown, including the main reference of the results  and putting also an arrow for the tendency observed in childhood (probably not different to the infancy after 3 years old)

6.- 3.1. The Mediterranean diet, is well described although I don´t find reported specific papers about with aging and changes on the microbiota that must be included and remarked and emphasized

7.-3.2. The Oriental diet. As the same orientation in this section although there is a reduced risk of several associated diseases with aging, there is not a clear benefit of health in aging people and the changes observed in the microbiota must be commented in more detail

8.-3.3. Effects on the microbiota by specific compounds in diet. Is OK.

This section is very well described in relation to the contents  of diet specifically related diet on fibers, fats and polyphenols

9.- Conclusions are clear and well explained

10.- References are well selected and appropriated

Author Response

RESPONSE TO REVIEWERS

 Resubmission of the manuscript with Reference ID biomedicines-890486

We thank the editor and the reviewers (for their critical review and valuable comments. We have taken into account all of their recommendations and suggestions. Itemized responses are listed below citing line number and exact change. All the modifications have been clearly highlighted using the “track changes” function in Microsoft Word, so that changes are easily visible to the editors and reviewers (See Revised Manuscript-Tracked changes).

REVIEWER 2

This is a very interesting review of an intriguing and very exciting problem of health related to the relationships existing between some components of the diets and the secondary effects on the intestinal microbiota in relation to the aging process.

I have included some small changes and suggestions in order to improve the manuscript

The references are very appropriate and in a good number included.

1.- I suggest a little change on your title by this other: “Effect of some diet components and intestinal microbiota composition on the achievement and maintenance of good healthy aging”

ANSWER: We thank the reviewer for his/her suggestion, but instead, we would like to change the title of the manuscript by this one: “RELATIONSHIP BETWEEN DIET, MICROBIOTA AND HEALTHY AGING”.

2.- The Abstract, is OK

3.- The Introduction, is OK

4.- 1.1 Microbiota, is OK

5.- 2. The interplay between aging and microbiota. Is OK

Figure 2 : In the different sections of the circle  components of different microbacteria populations, would be illustrative to include  the percentage number of every population in the diverse periods of life  shown, including the main reference of the results  and putting also an arrow for the tendency observed in childhood (probably not different to the infancy after 3 years old)

ANSWER: We thank the reviewer for his/her accuracy. As suggested, we modified Figure 2, adding the percentages of each bacteria population, the reference of the results and the tendency observed in childhood-adulthood. Please see the new version of the manuscript.

6.- 3.1. The Mediterranean diet, is well described although I don´t find reported specific papers about with aging and changes on the microbiota that must be included and remarked and emphasized

ANSWER: We agree with the reviewer, and we discussed in more detail the relationship between the Mediterranean Diet, aging and microbiota described in the reported papers. Please see the new version of the manuscript with tracked changes, lines 190-199.

Furthermore, recent data suggest that the MD promotes beneficial effects on the intestinal microbiota, favoring the diversity of the colon microbiota by increasing Bacteroidetes and Firmicutes and reducing Clostridium [37]. Another study carried out on 27 healthy subjects, who were monitored for MD compounds, showed how MD eating habits modified the composition and diversity of the microbiota. More specifically, they observed an increase in Bacteroidetes and a higher Bacteroidetes/Firmicutes ratio. This modification of the different microbiota populations has proven to exert anti-inflammatory effects. Accordingly, how MD can decrease the transcription factor E2F1 in colon cancer cells in order to decrease the proliferation of these cells has been studied. Additionally, it activates factors with protective effects against oxidative stress such as the transcription factor NRF2 [39,40].

7.-3.2. The Oriental diet. As the same orientation in this section although there is a reduced risk of several associated diseases with aging, there is not a clear benefit of health in aging people and the changes observed in the microbiota must be commented in more detail

ANSWER: We again agree with the reviewer, and we commented in more detail the relationship between the changes in the microbiota through Oriental Diet and aging. To this end, we introduced Curcumin, a typical Oriental Diet compound with proved impact on microbiota. Please see the new version of the manuscript with tracked changes, lines 289-295.

“Curcumin, a polyphenol found in turmeric root, is also abundantly present in OD. It has proven to be able of exerting anti-inflammatory and anti-tumor effects, among others [81,82]. Regarding its role in the microbiota, it has been seen to directly modify it. Since its absorption and bioavailability is low, it can be found in significant amounts in the gastrointestinal tract after its consumption. In fact, curcumin administration considerably changed the relationship between beneficial and pathogenic bacteria by increasing the abundance of Bifidobacteria, Lactobacili and reducing the loads of Prevotellaceae, Choriobacteria, Enterobacteriaceae and Enterococci [83].”

8.-3.3. Effects on the microbiota by specific compounds in diet. Is OK.

This section is very well described in relation to the contents of diet specifically related diet on fibers, fats and polyphenols

9.- Conclusions are clear and well explained

10.- References are well selected and appropriated